# Biofunctional Textiles for Aging Skin

**DOI:** 10.3390/biomedicines7030051

**Published:** 2019-07-17

**Authors:** Pierfrancesco Morganti, Gianluca Morganti, Claudia Colao

**Affiliations:** 1Dermatol Unit, Campania University, “Luigi Vanvitelli”, 80100 Naples, Italy; 2Dermatol Department, China Medical University, Shenyang 110001, China; 3ISCD Nanoscience Research Center, 00165 Rome, Italy

**Keywords:** skin barrier, skin redox, biofunctional textiles, chitin nanofibrils, nanolignin, air pollution, nanoparticulate, reactive oxygen species, environment, cosmeceuticals

## Abstract

The skin is the largest organ in the human body, acting as the first protective barrier against the external environment aggression, such as UV rays and atmospheric nanoparticulate pollutants. On the one hand, the skin employs different antioxidant agents to protect its natural oxidative balance. On the other hand, ageing phenomena are the main cause of skin barrier damages, leading to a disequilibrium in the physiological redox system. Thus, the necessity to find new innovative cosmetic means, such as biodegradable non-woven tissues able to load, carry and release active ingredients in the right skin layers. These innovative cosmetic tissues can not only protect the skin from toxic environmental agents, but may balance the natural skin barrier, also acting as anti-aging agents when their fibers are bound to the right ingredients. The proposed tissues, consisting of polysaccharide natural fibers made of chitin nanofibrils and nanochitin, seem to be an ideal candidate for the production of new and effective biofunctional textiles, also because they are able to mimic the skin’s extra cellular matrix (ECM) when electrospun. These innovative cosmeceuticals have shown the possibility of being used for food formulations as well as for topic anti-aging agents, having shown an interesting repairing effectiveness on skin and also on hair. Thus, they could be used both as active ingredient and as skin smart active carriers in substitution of normal emulsions, being also biodegradable, free of chemicals, and obtainable from waste material.

## 1. Introduction

The increasing world population, which reached 7.5 billion by 2015 and is projected to increase to 9.7 billion by 2050 and 11.2 billion in 2100 [1], is a consequence of the fertility decline and improvement of human survival [2]. The global population aged 60 years or over totaled 962 million in 2017—more than twice the amount in 1980—is expected to double again by 2050. Thus, globally, the number of persons aged 80 years and over is expected to increase more than threefold between 2017 and 2050, rising from 137 million to 425 million (Figure 1 and Figure 2).

In 2050, older people are expected to account for 35% of the population in Europe, 20% in North America, 25% in Latin America and the Caribbean, 24% in Asia, 23% in Oceania, and 9% in Africa. In conclusion, life expectancy worldwide is projected to rise from 70 years in 2010–2015 to 77 years in 2045–2050, and eventually to 83 years in 2095–2100 (Figure 2).

Inevitably, due to the ageing processes, body organs and tissues modify their structures, changing their normal physiological processes.

Thus, the skin loses its attractive and youthful appearance, showing the dreaded symptoms of wrinkles, blotches, dis-pigmentations, laxity, clogged pores, dry roughness, and tumor growths.

As a consequence, the quality of life of elderly people deteriorates through the loss of “stereotypical” physical attractiveness, reported as a “must” by the mass media’s magazine and televisions:" What is beautiful is good". According to Kligman [3], when evaluating the attractiveness of women and men within an elderly population, the unattractive tends to be viewed in terms of old age (i.e., deteriorated appearance, worn out, ugly).

By contrast, older people who have preserved their appearance age well are associated with youthfulness and personable qualities of being young and healthy.

In conclusion, unattractive people receive less help, are rejected, are thought to be morally or socially defective, and in general flawed in personality and worth [3].

To prevent to some degree these disadvantages, the role of cosmetics and aesthetic tools are considered very important in enhancing the more positive aspects of appearance, with or without possible plastic surgery interventions. This paper proposes to attempt improving the appearance of aging skin by using a smart carrier made of non-woven tissues that are embedded with antioxidant ingredients.

## 2. Skin

Skin, the largest human organ, represents 12–15% of body weight with a wifth of 0.5 m^2^ and a thickness of less than 3 mm [4,5]. The epidermis, with its stratum corneum (SC)*,* forms the outer covering of the body, creating a water and antioxidant barrier that is self-repairing and continually renewed.

Thus, it appears as the first line of defense from environmental aggressions, protecting the body from water loss and penetration by harmful agents, such as solar UV radiation and air nano-particulates (Figure 3 and Figure 4). Skin is therefore a complex and dynamic organ that is in contact with the environment, and as a consequence of its aggressive agents, undergoes aging. These aggressive agents are, in fact, the major source of free radical formation. They, aging at the level of cell membrane (Figure 5) of the extra cellular matrix’s components (ECM), can induce damages to polysaccharides, proteins, lipids, and DNA structures, being also accompanied by a decreased immunological surveillance [6].

The skin barrier consists, in fact, of specific proteins and lipids known as the brick and mortar of the epidermis [7]. Bricks are the dead cells corneocytes, which are filled with a large protein complex, known as the cornfield envelop. The space between these cells is filled with lipids that form the mortar and are attached to the cornfield envelop by calcium that seems to represent a key formation of the barrier which modifies its structure during lifetime with aging (Figure 5) [8,9].

Therefore, this lipid/protein scaffold determines both the barrier integrity and the antioxidant defense functionalities [10]. Beneath this barrier lies another thick layer of connective tissue, including the collagen-rich dermis and the underlying fatty layer, hypodermis.

However, skin becomes more vulnerable with age, when many naturally protective functions decrease: tissues show a slow and irreversible degeneration, evidenced first of all by fine lines and wrinkling of the face and neck (Figure 6).

Desire for an everlasting beauty plays, therefore, an important role in our society, meaning women and men are looking for treatments to prevent skin aging phenomena by the use of innovative rejuvenation products and procedures [9,10]. For example, a combination of the right photo-protective agents, by the use of sunscreens and synergistic co-antioxidants, such as vitamins E and C and other natural ingredients such as lutein, applied topically and contemporary taken by oral route [11]. Their activity, in fact, may help to protect the skin cell membrane, avoiding the damage to the SC’ lipids and proteins occurring during lifetime, because of the pro-oxidative activity of free radicals (Figure 4) and may as well help to stimulate hair stem cells, as also reported by our studies [12].

For this purpose, according to our recent experiments, the use of non-woven specialized tissues as basic carriers capable of releasing their entrapped active ingredients, could represent an innovative mean to prevent and ameliorate some of the negative effects that may strongly influence the apparent age of both women and men.

Naturally, these tissues are made of biodegradable and bio-polymeric scaffolds, organized as ECM (Figure 7) and able, therefore, to provide the physical and functional microenvironment in which cells live.

## 3. Skin Dressing

According to some authors, bio-functional non-woven tissues should be produced as innovative textiles made of biologically active materials characterized by their smart biological properties [12]. They, in fact, have to support the skin and prevent its aging processes, interacting with the skin in an intensive and smart manner, modulating for example superficial microbiota, the inflammation cascade, and the repairing processes of aged, wounded, or burned skin [12,14,15].

For this purpose and according to our recent experiences, the basic scaffolds, designed for these engineered tissues and closely mimicking the ECM architecture, seem able to respect the skin cells’ structure and function when adequately formulated (data not reported).

Non-woven tissues made of chitin nano-fibrils (CN), nano-lignin (LG), and biodegradable green polymers bound to antioxidant ingredients, therefore, seem to be the right solution to produce engineering scaffolds in which cells can grow, proliferate, and differentiate into a specific tissue during skin regeneration, thus showing their effectiveness to fight the effects of aging on the skin, evidenced by wrinkles and fine lines (Figure 6) [14,15] or by hair aging [12].

These CN-LG carrier tissues, in fact, have been organized as polymeric nano-composites which, twisted by natural nano-fibers and characterized by a high surface-to-volume ratio and a microporous structure, have shown not only to provide bio-functional and bio-structural support, but also to have the capacity of stimulating growth and regeneration capacity of living cells [12,13,14,15].

Due to their unique physicochemical properties, in fact, nano-fibers have not only superior stiffness as well as tensile strength and flexibility in surface functionalities, but also the same structure organization of ECM (Figure 7), thus playing a significant role in the transportation of bioactive molecules to the appropriate body sites [12,14,15,16,17,18]. This is also the reason why the in progress European Research Project PolyBioSkin is based on the use of natural polysaccharides, such as cellulose, chitin and starch, together with bio-polyesters, such as polylactic acid (PLA) and polyhydroxyalcanoates (PHA), to make biodegradable tissues for industrial production of innovative baby diapers, bio-active beauty masks and advanced medications [19,20,21,22,23].

## 4. Chitin Nano-Fibrils

Chitin is a glucose-based and nitrogen-containing polysaccharide, widely distributed in nature as the principal component of the exoskeleton of crustaceans and insects as well as of cell walls of some bacteria and fungi [24]. After cellulose (to which it is structurally identical), it is the second most abundant biopolymer on earth, representing a resource of raw material estimated at about 100 billion tons/year [24,25].

Chemically, chitin is a linear, poly-beta-(1,4)-*N*-acetyl-d-glucosamine with an acetamide group at C2 in place of the cellulose’ hydroxyl group (Figure 8). It occurs in nature as a highly ordered crystalline/amorphous structure made of micro/nanofibrils arranged in anti-parallel strands (Figure 9) [26,27]. Nanofibrils (CN) (also named nanocrystals or nanowhiskers) consist of rigid crystalline and flexible amorphous regions, which stack together to form layered and twisted structures, that give strong, loadbearing properties to the crustacean exoskeleton (Figure 10) [28,29]. The best chitin de-acetylated compound is named chitosan: the polymer is called chitin when it has an acetylation degree less than 50% with a mean nitrogen content of 6.5%, whereas it is named chitosan when its acetylation degree is of 60% and more, reaching up 9.5% nitrogen [30,31]. However, for its excellent mechanical properties and nanodimensions, CN has proven to represent an interesting filler for reinforcing natural polymer composites [31,32,33].

Its morphology and physicochemical characteristics, in fact, facilitate its adhesion and dispersion in polymeric matrices, influencing the stress transfer into these nanocomposites [26,27,28,29]. Moreover, due to their bio-compostability, non-toxicity, biodegradability, biological activities, and tissue-film-forming capacities (characterized by their ECM-like structure when electrospun), chitin and chitosan are widely made use of in medicine, in cosmetics and in the food industry [14,15,16,17].

## 5. Conclusions

As previously reported, the primary function of skin is to act as a barrier against material being transferred in both directions, from outside to inside and vice versa. This important organ, in fact, interacts continuously with its environment, leading to situations affecting the physiological homeostasis of skin and the whole body. Thus, the necessity to find biodegradable and biocompatible carriers able not only to protect the skin from environmental aggressions, but also to temporarily load and carry designed and selected active ingredients, in order to release them in controlled doses at a chosen level of its different layers [34,35,36]. The proposed tissue carriers, made of biodegradable sugar-like biopolymers reinforced and oriented by nano-chitin, seem to be an ideal candidate to support cell adhesion and proliferation necessary for tissue engineering and regeneration [34,35,36,37]. Their fibers, in fact, according to studies of our research group, realized both in vitro and in vivo, have shown to possess excellent mechanical properties, and the ability and flexibility to protect skin from the environmental aggression [31,32,33,34,35], together with porosity and pore interconnectivity, achieving a proper skin response with antibacterial, anti-inflammatory, antioxidant, and skin repairing effectiveness, without producing any adverse effects [36,37,38,39,40,41].

These tissues, made of polysaccharide-natural fibers obtained from waste materials, may be used as innovative carriers for the manufacturing of different cosmetic products, characterized by the proper and selected active ingredients they contain. Thus, for example, it will be possible to produce innovative anti-aging beauty masks (Figure 11) bounding for example antioxidants and immunomodulating compounds to the tissue fibers. With this new technology, it will be possible to produce innovative and smart cosmeceuticals without the use of preservatives, emulsifiers, and other chemical agents, as previously reported by the PolyBioSkin project. These tissues, in fact, could be made through electrospinning or casting technologies, and if packed and distributed in a dry state, they may be free of water and any chemicals such as preservatives and/or emulsifiers, fragrances, colors, and other chemicals often causing allergic or sensitizing reactions.

These innovative cosmetics, made of biodegradable and skin-friendly tissues, when bound to the right active ingredients and applied painlessly, gently, and directly on the skin as a clothing, are able to provide in about 30 minutes a fast-repairing and rejuvenation effect. Moreover, as these ingredients would be obtained from waste materials, they may reduce the formation of CO2, saving both health and the environment [41].

This is the goal of our in-progress studies, by which we are trying to move the production of biodegradable tissue carriers from the lab stage to large-scale manufacturing.

## Figures and Tables

**Figure 1 biomedicines-07-00051-f001:**
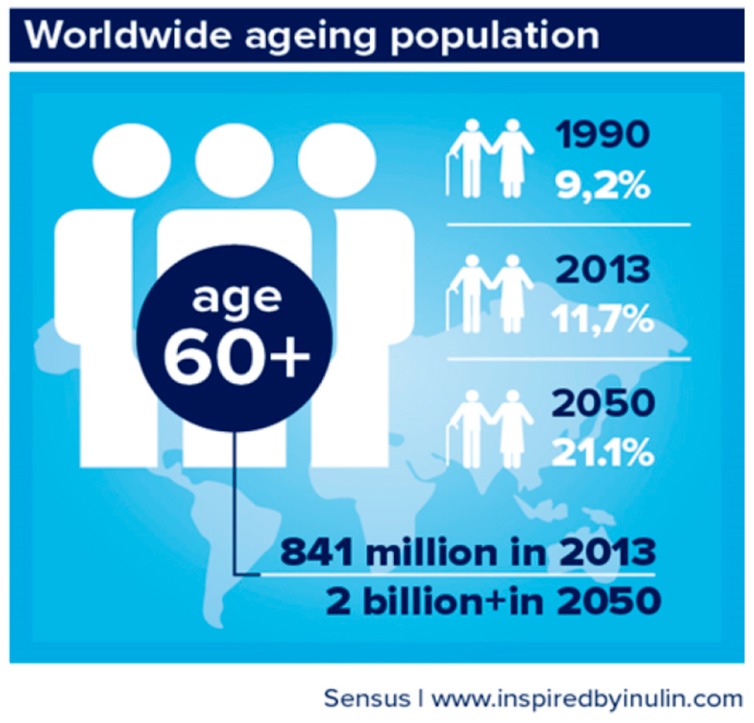
The aging population worldwide.

**Figure 2 biomedicines-07-00051-f002:**
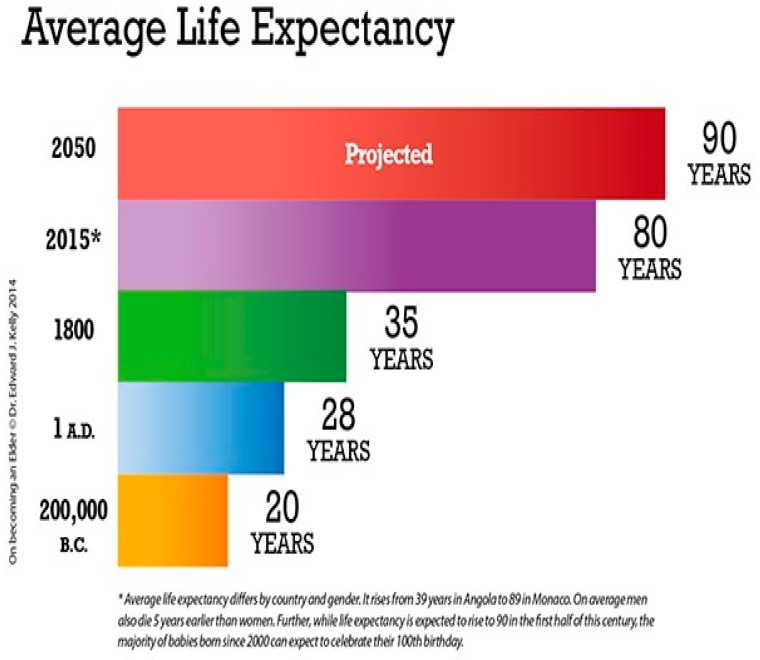
Average life expectancy.

**Figure 3 biomedicines-07-00051-f003:**
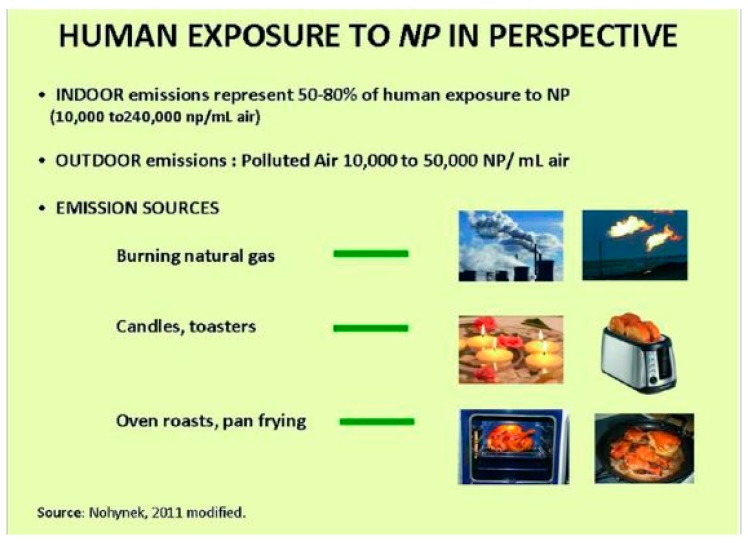
Indoor and outdoor particulates exposure.

**Figure 4 biomedicines-07-00051-f004:**
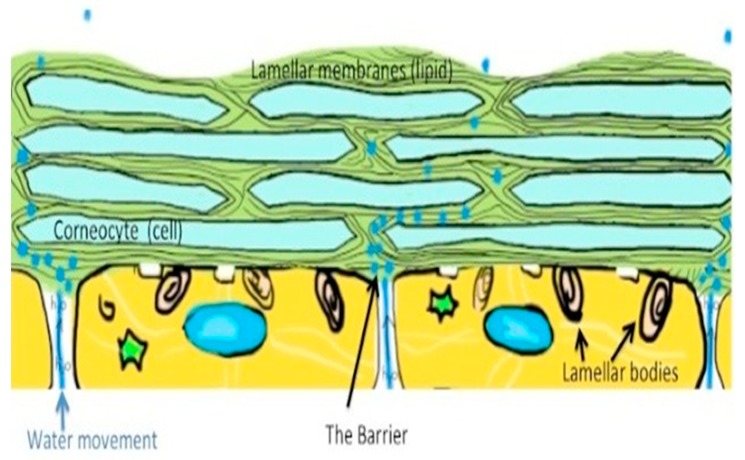
The so-called skin mortar and brick: lipids are the mortar and corneocytes are the bricks [13].

**Figure 5 biomedicines-07-00051-f005:**
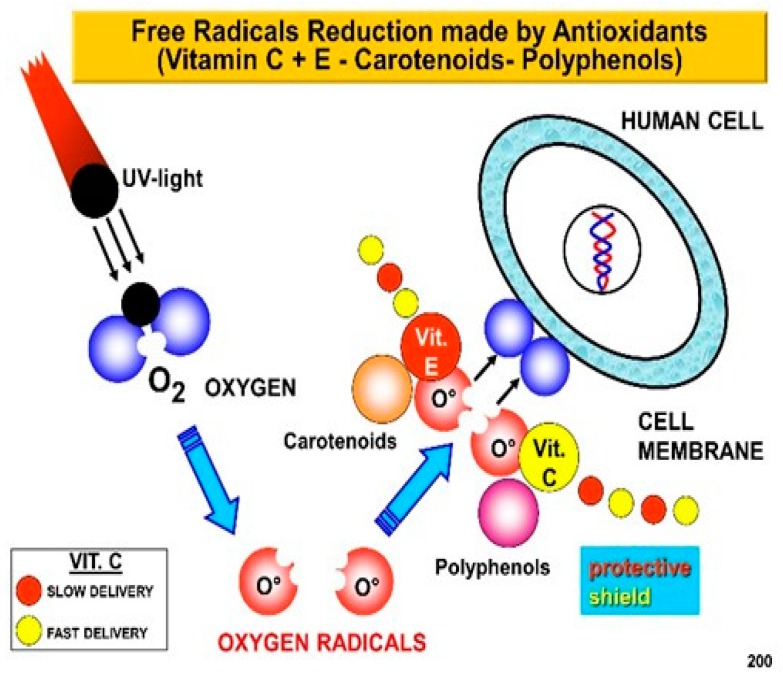
Free radical activity and antioxidants effectiveness at cell level.

**Figure 6 biomedicines-07-00051-f006:**
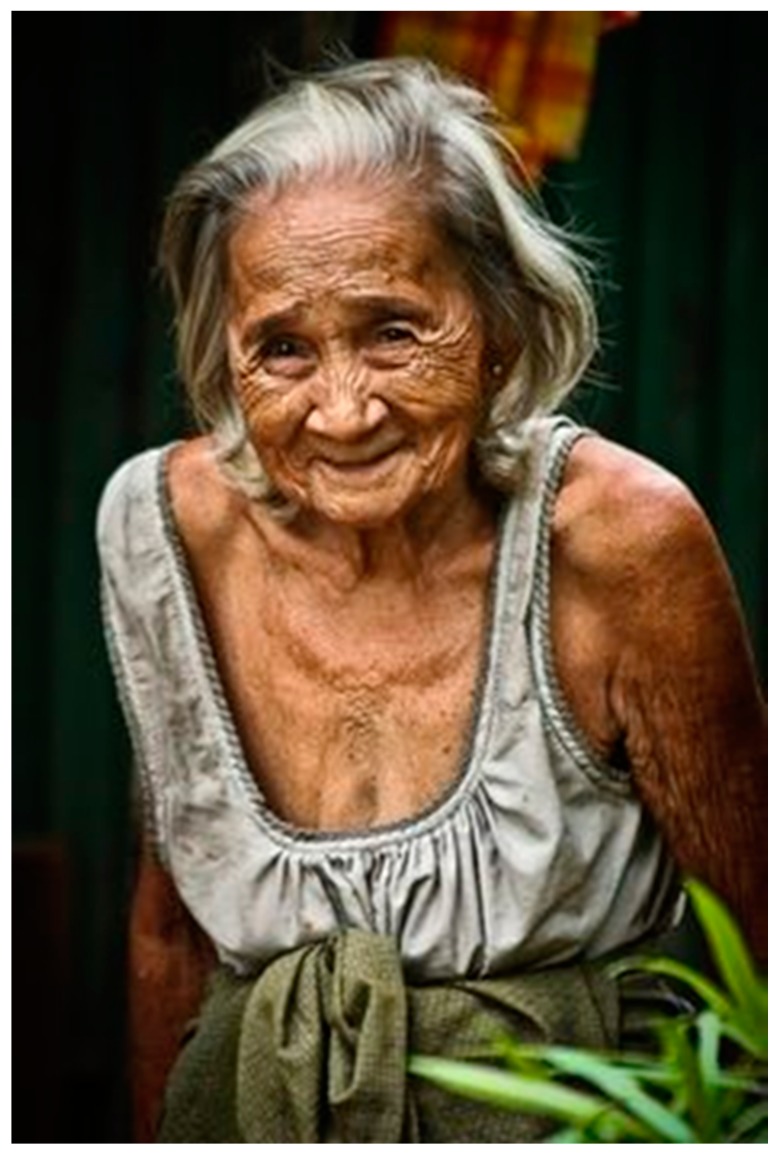
Aged skin.

**Figure 7 biomedicines-07-00051-f007:**
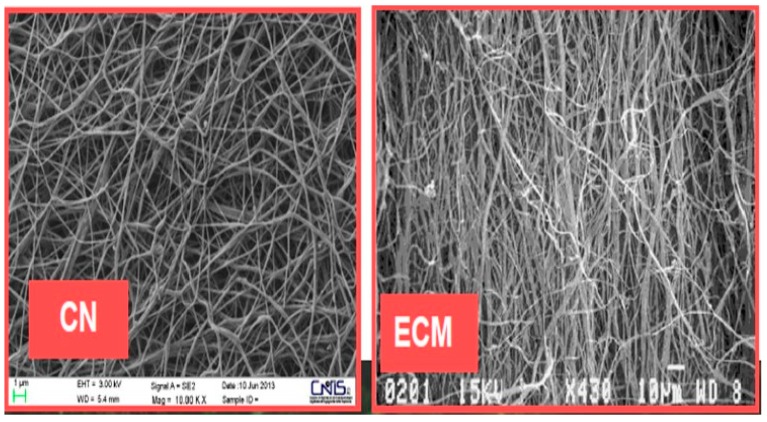
The CN-tissue structure at SEM (left) compared to natural ECM (right).

**Figure 8 biomedicines-07-00051-f008:**
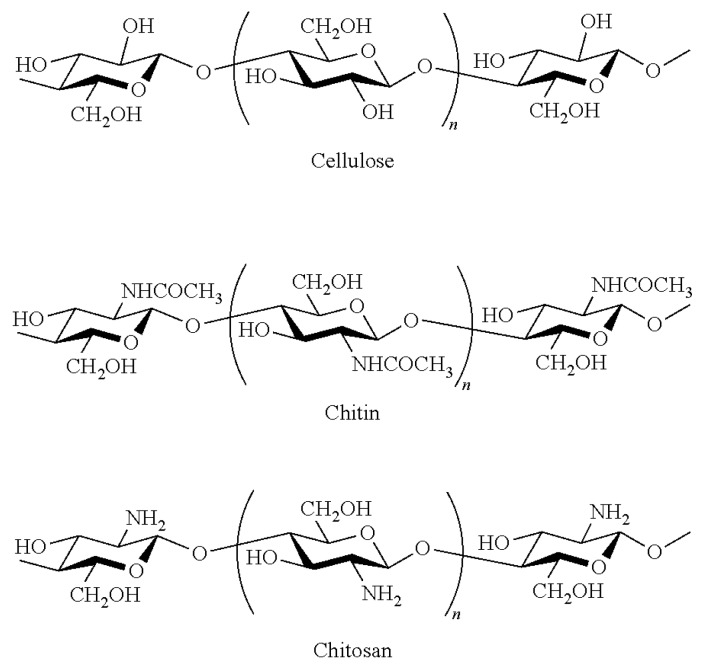
Chitin, Chitosan, and Cellulose chemical formula.

**Figure 9 biomedicines-07-00051-f009:**
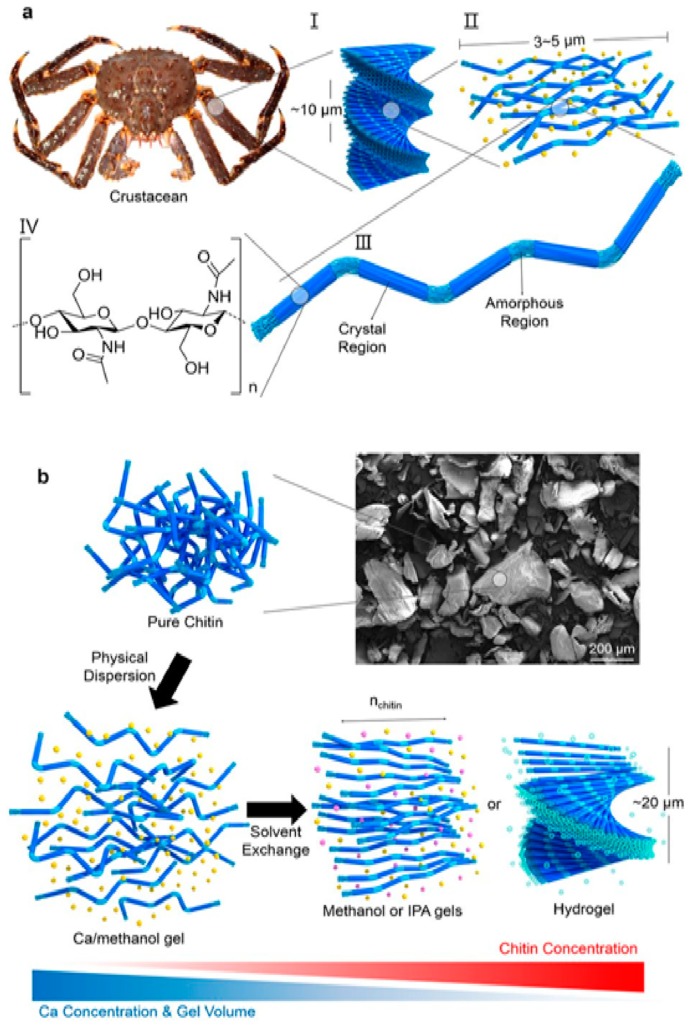
Crystallin and amorfous structure of chitin (courtesy of Oh et al. [29]).

**Figure 10 biomedicines-07-00051-f010:**
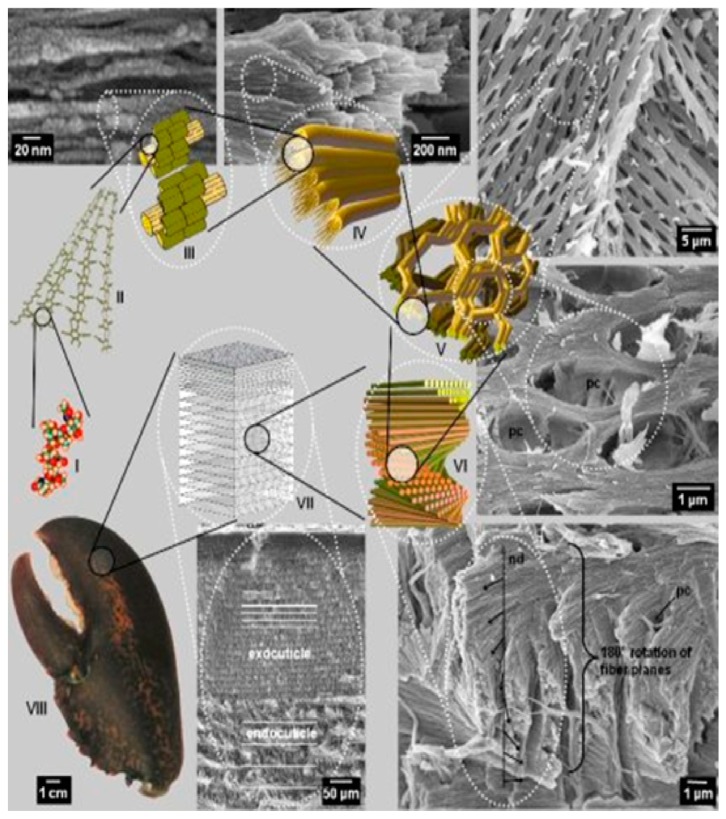
Chitin Nanofibril arrangements (courtesy of Fabrius et al. [30]).

**Figure 11 biomedicines-07-00051-f011:**
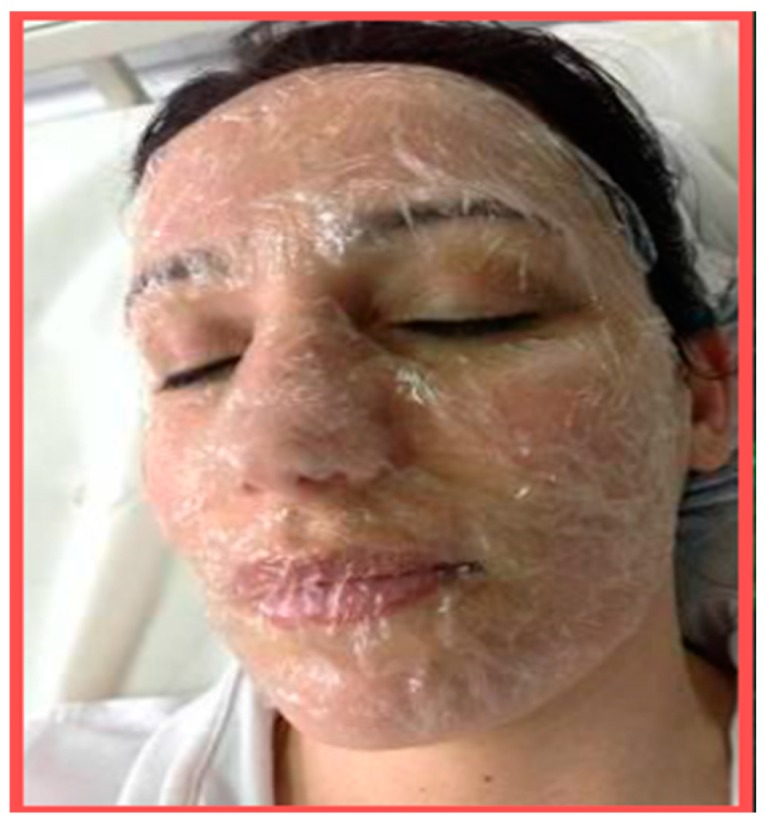
Chitin nanofibril-based beauty mask made by casting technology.

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
