# Peer review of "Biofunctional Textiles for Aging Skin"

_biomedicines, 2019, doi:10.3390/biomedicines7030051_

Round 1
Reviewer 1 Report
The paper has been improved by the firt time submitted. However I consider that the sections: Intriduction and skin coild be improved. General english gramar corrections should be considered. All figures have been sent twice, must modify this. Abstract should be carefully rewritten.
Author Response
Both introduction and skin coild have been improved, general english gramar corrections were done and abstract was carefully modified.
Reviewer 2 Report
No more comments
Author Response
Thanks.
Reviewer 3 Report
Dear authors,
I am afraid the previous comments are not addressed properly
Author Response
Authors’ response
1,2 . This summary has been improved as well the figure qualities
3. FIG 4 is part of my personal file
4,5,6,8. The bio activity of Chitin has been reported regarding only its topic activity because this paper is considered a short communication.
Moreover, something about the use of CN on aged hair has been reported with a phrase and inserted into the references.
7. The Skin scaffold activity of CN has been reported from other papers of our group.
Round 2
Reviewer 3 Report
Dear author,
I am afraid if the paper can be published as an article.
Either change into short communication or something similar
This manuscript is a resubmission of an earlier submission. The following is a list of the peer review reports and author responses from that submission.
Round 1
Reviewer 1 Report
Dear author
The research article submitted under the title" Biofunctional Textiles For Aged Skin" by Morganti P et al. to journal of Biomedicines has been evaluated for its composition, quality and significance. This paper describes the applicability of biodegradable non-woven tissues that can protect the skin from the hazard environment and to equilibrate the skin to the nature. For this, the author(s) implement chitin as biofunctional textile tissues.
However, I would like to raise the following questions.
Therefore, I suggested its publication in Biomedicines after the mandatory revisions, as following;
1) The abstract should be improved.
2) The quality of the figures (Fig 1, 2, 3, 8) is low. Should be improved
3) Figure 4 doesnot indicate whether the figure is the author(s) or come from other sources (should be clear---of course depends on the journal guidelines).
4) The bioactivities of the chitin should be reviewed (as CNFs are more potential for potent functional foods for various diseases)
5) Chitin nanofibers (CNFs) have hair growth promoting activities—this should be reviewed as the product supposed to be used for aged skin
6) The UV protective effect of chitin should be included in the study
7) Chitin scaffolds -----include in the review
8) Overall, I am in doubt if the article can be considered as a review paper (too short to be considered as a review paper---even too few references)
Reviewer 2 Report
Major issue : I agree that their biofunctional textiles have a beneficial effect on the healing of burned or wounded skin. In the burned or wounded skin, those may act as an ECM because the epidermis is damaged and then the dermis is exposed. However, I can not agree that those will be effective for aging skin. Aged skin has an intact stratum corneum. Therefore, active ingredients within biofunctional textiles can not penetrate through stratum corneum.
Minor : I write them in corrected pdf file.

Reviewer 3 Report
The presented manuscritp is written to be a review of biofunctional textiles for aged skin.
I really consider the en englis grammas and expresions should be carefully corrected along all the manuscritp.
Pay especial attention on the figures presented, must be adecuate for being publicated on a scientific journal and must really clarify the cited text. Examples of figures which could be improved number 4 and 6, but reconsider well all of them
Better explanation of the biofuntional textiles should be given
Conclusion section are not in agreement with the previous sections. May be some of the conclusions should be better explained before?
As being a review for biofunctional textiles for aged skin, perhaphs more real information of all the different biofunctional textiles present in the scientific field should be descrbied not only the chitosan ones?
Author Response
1) Conclusion section are not in agreement with the previous sections. May be some of the conclusions should be better explained before?
1) answer: we described better some of conclusions
2) As being a review for biofunctional textiles for aged skin, perhaphs more real information of all the different biofunctional textiles present in the scientific field should be descrbied not only the chitosan ones?
2) answer: we included other real information